# Establishment of a Stable BK Polyomavirus-Secreting Cell Line: Characterization of Viral Genome Integration and Replication Dynamics Through Comprehensive Analysis

**DOI:** 10.3390/ijms26125745

**Published:** 2025-06-15

**Authors:** Tamara Löwenstern, David Vecsei, David Horner, Robert Strassl, Anil Bozdogan, Michael Eder, Franco Laccone, Markus Hengstschläger, Farsad Eskandary, Ludwig Wagner

**Affiliations:** 1Institute of Medical Genetics, Center for Pathobiochemistry and Genetics, Medical University of Vienna, 1090 Vienna, Austria; tamara.loewenstern@meduniwien.ac.at (T.L.); david.horner@meduniwien.ac.at (D.H.); franco.laccone@meduniwien.ac.at (F.L.);; 2Division of Nephrology and Dialysis, Department of Internal Medicine III, Medical University of Vienna, 1090 Vienna, Austria; david.vecsei@meduniwien.ac.at (D.V.); michael.eder@meduniwien.ac.at (M.E.); 3Division of Clinical Virology, Department of Laboratory Medicine, Medical University of Vienna, 1090 Vienna, Austria; robert.strassl@meduniwien.ac.at (R.S.); anil.bozdogan@meduniwien.ac.at (A.B.)

**Keywords:** BK polyomavirus, genome integration, cell line, rearrangement, NCCR, non-coding control region, VP1, LTAg

## Abstract

Polyomaviruses have the potential to cause significant morbidity not only in transplant medicine, but also in other forms of disease or variants of immunosuppression. In kidney transplant recipients or recipients of human stem cell transplants, the BK-Virus is the major proponent of manifestations such as BKPyV-associated nephropathy or hemorrhagic cystitis. As no polyomavirus-specific drug with proven in vivo effects has been developed so far, methods to screen for such drugs are important. This work describes the establishment of a virus-secreting cell line. By infecting a pre-established monkey kidney cell line (COS-1) with a non-rearranged human BK polyomavirus isolated from a kidney transplant patient suffering from BKPyV-associated nephropathy, a continuously replicating cell type with consistent virus secretion could be established and was termed COSSA. Measurements of BKPyV replication, virion production, and secretion were performed both intracellularly and in the cell supernatant. Viral proteins such as VP1 and LTAg were accurately tracked by confocal microscopy, as well as by immunoblot and qPCR. An intracellular flow cytometry (FACS) assay detecting VP1 protein was established and revealed an expanded range of positive intracellular signals. The viruses produced proved to be infectious in human tubular epithelial cell lines. Long-range sequencing of the COSSA genome using Oxford Nanopore Technology revealed a total of five distinct BKPyV integration events. One integration of a partial BKPyV genome was located upstream of the epidermal growth factor receptor gene. The second and third, both truncated forms of integration, were close to histocompatibility gene locuses, while the fourth was characterized by a ninefold and the fifth by a fourfold tandem repeat of the BKPyV genome. From both of the repeat forms, virus replicates were derived showing deletions/duplications on early and late genes and inversions within the non-coding control region (NCCR). This pattern of repetitive viral genome integration is a potential key driver of enhanced viral replication and increased virion assembly, ultimately supporting efficient virus egress. Quantitative PCR analysis confirmed the release of approximately 10^8^/mL viral units per 48 h from 2 × 10^5^ COSSA cells into the culture supernatant. Notably, the NCCR region of the most frequent copies of circular virus and the integrated tetrameric tandem repeat exhibited a rearranged configuration, which may contribute to the observed high replication dynamics. The establishment of a consistent methodology to generate and secrete BKPyV from a cell line is expected to significantly facilitate antiviral drug development.

## 1. Introduction

The BK polyomavirus (BKPyV) usually remains in a latent state in immunocompetent individuals after infection during childhood. However, it is reactivated in patients undergoing immunosuppression, particularly in kidney transplant recipients [1,2,3]. Under such conditions, it can produce high viral copy numbers that appear in the urine and in the blood [4,5]. It was originally assumed that the virus persists episomally at the site of its tropism, which for BKPyV is the tubular epithelia in kidneys and the urothelia [6]. This view has recently broadened significantly with the advent of second and third generation sequencing and the new method of BKPyV capture sequencing [7,8].

The BKPyV genome comprises 5140 bp, which codes for an early transcribed region, the late transcribed genes, and the regulatory non-coding control region (NCCR). The virus uses the host’s enzymes to replicate its genome and to transcribe and translate its proteins, which is responsible for virion formation. Due to this high dependency on the enzymes of the mammalian host cell, already established, highly effective virustatics that target different viral proteins involved in virus replication, such as the HIV virus or Hepatitis C virus, are not effective for the BKPyV polyomaviruses [9].

Various compounds have been tested in clinical studies, but with no convincing results [10,11,12]. Others were biochemically tested [13] or suggested to have an in vitro virustatic effect against polyomaviruses [14,15,16,17].

To make progress in this area, virus-secreting cell culture models and cell lines provide a platform for drug discovery, as they represent a consistent and controllable source of viral particles [18,19,20]. Cell lines enable rapid screening of antiviral agents by monitoring the reduction of viral secretion or replication [21]. High-throughput assays can be used to measure the effect of drug candidates on viral load in the culture supernatant [22].

By using virus-secreting cell lines, the mechanisms by which compounds interfere with viral replication, assembly, or exit can be elucidated. A critical development was the generation of COS cell lines, achieved by transfecting the parental African green monkey kidney CV-1 cell line with a replication-incapable SV40 genome. These modified cell lines are incapable of producing infectious SV40 particles [23]. However, subsequent studies revealed that COS cells remain permissive to replication-defective SV40 mutants, thus enabling their replication and maintenance within the cells and thereby supporting their propagation.

Virus-secreting cell lines are indispensable in preclinical research for assessing the efficacy, safety, and mechanism of action of antiviral drugs, ultimately aiding in the development of effective therapeutic strategies.

For the above-mentioned reasons, we have worked on establishing a BKPyV secreting cell line on basis of the COS monkey kidney cell line by infecting it with the BKPyV secreted from cultured human decoy cells, which did not show any rearrangement. For evaluating BKPyV replication and shedding, we systematically assessed multiple methodologies, including immunofluorescence, flow cytometry (FACS), immunoblotting, and quantitative PCR (qPCR), to determine their suitability for drug screening applications. In addition, we identified the integration sites of BKPyV within the monkey cell genome and characterized the genomic structure of the virus originally used for infection by employing Oxford Nanopore Technologies (ONT) long-read sequencing. The establishment of a consistent methodology for generation of BKPyV production and secretion from a cell line is expected to aid in drug repurposing and screening/development of novel inhibitory compounds.

## 2. Results

### 2.1. Characterization of the COSSA Cell Line

The monkey kidney cell line COS-1 was infected with BK polyomavirus (BKPyV), isolated from the supernatant of decoy cells obtained from a kidney transplant recipient. By day 5 post-infection, extensive cytopathic effects were observed, and the majority of cells detached and died. A small fraction of adherent cells survived and formed colonies when maintained in fresh culture medium. These surviving cell colonies were first passaged on day 12. By the third passage, it became evident that a stable cell line had emerged, characterized by constitutive production and secretion of BKPyV. Subcloning of this cell line resulted in the establishment of multiple, nearly identical BKPyV-producing clones. The infected COS-1-derived line was designated COSSA.

To evaluate the efficacy of intracellular BKPyV replication and virion assembly, an immunoblot assay was established to assess the presence of the BKPyV VP1, the major capsid protein encoded by late-phase viral genes. VP1 serves as the principal structural component of the viral capsid, playing a crucial role in virion formation. The established COSSA cell line exhibited high expression of the target protein by immunoblot (Appendix A).

### 2.2. Polyomavirus Protein Confocal Immunofluorescence LTAg and VP1 in COSSA

To evaluate BKPyV capsid protein VP1 expression in infected COSSA cells, we employed a BKPyV VP1-specific monoclonal antibody to assess both the localization and intensity of VP1 expression at the single-cell level. Confocal immunofluorescence analysis revealed VP1 along with LTAg expression. As expected, LTAg was produced by all cells. Additionally, most cells exhibited VP1 signals of varying intensity, indicating active progression of BKPyV infection (Figure 1A).

### 2.3. BKPyV-VP1 Protein Measurement and Cell Cycle Analysis by Intracellular FACS in COSSA Versus COS-WT

To determine intracellular staining of virions via VP1 with the anti-VP1 mAb 4942, fixation and permeabilization with 70% ethanol resulted in VP1-specific intracellular staining by FACS. The mean fluorescence intensity of infected COSSA cells was significantly higher compared to that of parental COS cells. Statistical analysis was performed using Mann–Whitney U test (n = 4, *p* = 0.021) (Figure 1B,C).

To assess whether BKPyV infection alters the DNA replicative behavior of the cell line, we evaluated and compared the cell cycle profiles of COSSA and parental COS cells. Cell cycle analysis comparing the COS and COSSA cell lines revealed a higher proportion of cells in the mitotic polyploid phase in COSSA 24 h post-passaging. However, this difference did not reach statistical significance (two-sided unpaired Student’s *t*-test, n = 6, *p* = 0.1648; Figure 1D).

### 2.4. Infectivity Analysis of COSSA-Secreted BKPyV in Human Tubular Progenitor Cell Lines

To test whether the COSSA-secreted virus particles were able to infect human renal tubular progenitor cells lines, these cell lines were prepared from the urine of three different kidney transplant patients (Figure 2AI,BI,CI). As described in the methods sections [24], the cell lines were exposed to 1 × 10^8^ copies/mL for three days and further grown in the REBM/REGM cell culture media. Under culture conditions before infection (Figure 2AI(A,B)), no vacuolation was observed. As a control, the same tubular progenitor cell line was maintained in parallel under identical conditions but without exposure to infectious BKPyV particles (Figure 2AI(C)), and no vacuolation appeared. The progression of viral titer after infection was observed in tissue culture on day 20, when about 5% of the cells showed lipoid degenerative vacuoles, and on days 26 (Figure 2AI(D,E), when 50% of the cells showed lipoid degenerative vacuols. On day 34, control cultures showed no vacuolation (Figure 2AI(F)), whereas in the infected cultures (Figure 2AI(G,H)), more than 90% of the cells had detached, with only a small proportion remaining adherent to the tissue culture plate. Such vacuolation was also observed in biopsy sections of BKPyVAN [25] and in vacuole-like vesicles of BKPyV infection of bladder microvascular endothelial cells [26]. Cell morphology in H&E staging shows that most cells have apoptotic condensed chromatin and disrupted cell morphology, as shown in (Figure 2AI(H)).

Beginning with the onset of initial vacuolation, the virus titer increased from 10^4^ (day 20) to 10^9^ (day 26), 10^10^ (day 30), and, at the time of cell lysis, to 10^11^ (day 34) (Figure 2BI). At day 26 post-infection, 79% of the tubular epithelial progenitor cells were positive for VP1, displaying variable staining intensities and subcellular localization patterns. By day 30, over 95% of the cells exhibited VP1 positivity, as shown in Figure 2CI.

### 2.5. BKPyV Protein Confocal Immunofluorescence VP1 and AQP1, a Membrane Marker Protein of Infected Tubular Progenitor Cell Lines #2 and #3

To evaluate the intracellular presence of BKPyV, the BKPyV VP1-specific monoclonal antibody was used on day 24 after infection with COSSA-derived virus particles. This was performed in two different tubular progenitor cell lines, #2 and #3, and stained in combination with aquaporin-1 (AQP1) using dual-color confocal microscopy. Both cell lines exhibited strong VP1-specific immunoreactivity (Figure 2CI, upper and lower panel), and the presence of intracellular lipoid vacuoles was evident, as shown in Figure 2CI (upper panel, arrow).

### 2.6. BKPyV Genome Integration into the Host Cell Genome

As a final step, we investigated whether and where the integration of BKPyV into the host genome had taken place in the COSSA cell line. For this reason, we isolated DNA from COSSA cells and sequenced it using the ONT Nanopore platform. As a result, BKPyV genome integration was found at five different sites. Of interest is the integration upstream of the EGFR gene, as shown in the (Appendix A), but at this site, the BKPyV genome could not be identified in its entirety. Only fragments of capsid proteins, the non-coding control region (NCCR), and a fragment of the LTAg protein are contained (contig 1, as shown in Table 1). A second site of integration was identified in the vicinity of a histocompatibility gene locus of the host cell. Only the leading 1491 bp of the BKPyV genome were identified in contig 2 (Table 1), where the NCCR, the agnoprotein, and a fragment of VP2 and VP3 were contained; in contrast, the coding part of VP1 and LTAg was missing (Appendix A). At another major histocompatibility gene locus, 1668 bp of the BKPyV genome, comprising the anterior part of the LTAg gene and NCCR, were found integrated with contig 3 (Appendix A). In addition, we identified a large, repetitive BKPyV in contig 4 encoding the BKPyV genome in nine tandem repeats spanning over 45,481 bp (Figure 3A and Appendix A), but each of the BKPyV genomes lack some internal part of the LTAg gene, including the origin binding domain (OBD) and the helicase domain; despite this deletion, the reading frame remains intact following splicing and results in a protein as a truncation product (Figure 3B). An internally truncated LTAg protein variant is generated through alternative splicing, in which exon 1 is joined to both the proximal and distal segments of exon 2, resulting in a transcript that encodes a 287-amino-acid polypeptide. At the protein level, this variant retains the N-terminal region, including the J domain and the conserved LXCXE motif, which mediates retinoblastoma (Rb) protein interaction. Additionally, it includes a 95-amino-acid region corresponding to the host interaction domain, as illustrated in Figure 3B. A schematic representation of the coding regions within the circular minichromosome is provided below. Furthermore and most importantly, a large repetitive BKPyV contig 5 encoding the entire BKPyV genome was identified, including the NCCR, which was organized in a fourfold tandem repeat comprising 20,460 bp (Figure 3C and Appendix A). This structure provided three times the sequence of the BKPyV genome and a fourth in a truncated version. Sequence alignment of the encoded VP1 protein reveals that the viral isolate clusters with genotype I reference strains, indicating that it belongs to BKPyV genotype I.

Sequence analysis of the non-coding control region (NCCR) revealed a structural rearrangement in which the initial portion undergoes an inversion of the OPQ region, following the S segment, within the first three tandem repeats of the BKPyV genome (Figure 3D). The blocks of the NCCR are depicted in accordance with earlier authors [27,28]. It is hypothesized in accordance with the work of earlier authors that such a structural modification, named rr-NCCR, may contribute to the enhanced replication efficiency of the viral genome by repeated enhancers and transcription factor binding sites, such as SP1 [29,30]. Indeed, the most frequently identified circularized BKPyV read originated from the replication of the third tandem repeat, as demonstrated in the subsequent analysis further below. The fourth repeat is truncated.

### 2.7. SV40 Genome Fragments Incorporated into the COS Cell Genome

The COS cell line was originally established by infection with the SV40 virus [23]. However, it was found that no mature SV40 viruses were produced or secreted in the cell line. Nevertheless, it was found that the SV40 LTAg is expressed as a protein [31]. For this reason, reads containing SV40 genome insertions were searched for in the ONT sequencing dataset. As expected, such insertions were identified. In contig A (Table 1, Figure 3E), an integrated SV40 fragment was detected in the COSSA genome. This 4124 bp insertion comprises the complete coding sequence for the SV40 large T antigen (LTAg), the non-coding control region (NCCR), the agnoprotein coding region, and a portion of the coding sequences for VP2 and VP3. A further integration site of 1560 bp was identified in contig B (Figure 3F), encompassing partial coding regions of VP1, VP2, and VP3. Notably, this integrated sequence lacks any promoter elements.

### 2.8. Identification and Structural Analysis of Non-Monkey Mapped BKPyV-Specific Reads

In order to elucidate the genome organization of the replicating BKPyV prepared for egress, and motivated by the observed variability in integrated genomes, notably that the largest 45,481 bp (contig 4) tandem repeat lacks the rr-NCCR seen at the tetrameric tandem repeat (contig 5), we looked for BKPyV genome replicates that do not map to the monkey genome. These sequences likely represent BKPyV genomes that have undergone rolling cycle replication and are primed for packaging into virions prior to egress.

The analysis of BKPyV-specific reads that did not align to the monkey genome revealed a broader distribution of replication variants. However, two predominant variants were identified: one comprising 300 reads with a genome size of 4820 bp (Figure 4A) along with an additional subvariant of 4815 bp (Figure 4B), and a third comprising 200 reads with a size of 4548 bp (Figure 4C) A histogram of the sequence length distribution of BKPyV-specific reads is shown in Appendix A. Notably, reads of increased sequence length (beyond 8000 bp) appear to correspond to tandem repeats of the dominant sequence peaks, as illustrated in the inset of Appendix A.

The alignment of the reads against the concatenated contigs confirmed a circular genomic structure, consistent with the known morphology of the BKPyV minichromosome. The full read assembly is provided in the Appendix A. Notably, the circular structure frequently exhibited fragmentation within the NCCR region in the ONT read, accounting for approximately 30% of the identified reads in the IGV pile-up (Appendix A).

Of particular interest, the 4820 bp variant (Figure 4A and Appendix A) exhibits the same NCCR inversion pattern observed in the tetra-BKPyV repeat identified in contig 5 (Table 1 and Figure 3D) and corresponds precisely to the integration contig mapped between positions 11,508 and 16,331 bp. It is interesting to note that this non-mapped read is identical in sequence to the formerly mentioned part of contig 5, containing a duplication of the first 138 bp of the NCCR, but as an inversion at the 5′ of the genuine NCCR. This inversion contains the ORI together with enhancers and transcription factor binding sites and therefore appears twice as a most interesting novel form of rr-NCCR (shown in red in Figure 4A). In addition, this variant has a repeat of translocated bp 4860–5141 at the site where the coding region of the agnoprotein was deleted. In general, coding was found highly similar to the reference strain 12B-2 with the accession NrAB301099.1.

The second most frequently identified circular read (circular contig 2) measured 4815 bp (Figure 4B). This variant exhibited a repeat and translocated NCCR together with an inversion of the region comprising bp 4454–5145 inserted into the coding region of the LTAg protein and a duplication for small TAg coding. However, alternate splicing of the LTAg gene fragments might code for a truncated but partly functional LTAg protein, as outlined in Figure 3B. The correct coding sequences for the structural proteins, including agnoprotein, VP1, VP2, and VP3, are preserved (Figure 4B and Appendix A). It was derived from the ninefold BKPyV tandem repeat within contig 4, which spans 45,481 bp and comprises nine consecutive BKPyV genome units (Figure 3A). The altered LTAg protein sequence is depicted in Figure 3B. The accurate site to which the circularized read starts is at 39,673 and ends with 44,487, which represents the ninth BKPyV genome repeat within contig 4, as shown in Figure 3A and Appendix A.

The third circular BKPyV read, measuring 4545 bp (Figure 4C), might originate or have some relation to the third repeat within the tetrameric BKPyV tandem repeat (contig 5), as illustrated in Figure 3C, Appendix A. Its genomic arrangement differs from the circular read depicted in Figure 4A, as it harbors a deletion within the structural proteins agnoprotein, VP2, and VP3. The coding sequence aligns with nucleotide positions 11,119–15,547 and 15,604–15,920. The decoding of circular contig 3 initiates 389 bp upstream relative to circular contig 1. Additionally, the deletion spanning nucleotides 15,547–15,604, while otherwise preserving homology with contig 5, suggests that this tetrameric tandem repeat serves as the origin of this circular repeat. However, it may fulfill a distinct biological function.

### 2.9. Genomic Characterization of the BKPyV Clinical Isolate Used for COS-WT Cell Infection

To determine whether the BKPyV-infecting COSSA cell line originated from a single viral clone or a heterogeneous viral population, viral DNA from the used inoculum was extracted separately and subjected to Oxford Nanopore Technologies (ONT) long-read sequencing. The analysis identified a canonical BKPyV genome of 5141 base pairs, harboring nucleotide substitutions at four distinct positions, indicating the presence of minor mutated variants within the initial viral population (shown in Appendix A). These findings clearly indicate the presence of a heterogeneous viral population comprising mutated variants, none of which exhibited genomic rearrangements such as rrNCCR, as described above for COSSA. The full BKPyV sequence and the identified nucleotide variants are presented in the Appendix A, along with the corresponding IGV pile-up visualization, which is shown in the Appendix A.

## 3. Discussion

In this study, we aimed to establish a cell line that continuously secretes BK polyomavirus (BKPyV) while maintaining growth. We successfully generated such a cell line by infecting the COS monkey kidney cell line with a BKPyV isolate from a patient with polyomavirus-associated nephropathy (PyVAN). This cell line produces high titers of infectious virions that are released into the culture supernatant, and this phenotype has been stably maintained over more than 50 passages. To investigate the genetic basis of this persistent viral secretion, we performed whole-genome sequencing using Oxford Nanopore Technology (ONT). Our analysis identified the integration of the BKPyV genome at a minimum of five distinct loci within the host genome. Additionally, at two of these integration sites, the complete viral genome was detected as a fourfold and a ninefold tandem repeat, respectively, providing insights into the structural organization of the integrated sequences. In addition, circular reads were found that are in relation or decoded from the integration sites and show diverse structural variation (duplications, deletions, and inversions). However, taken together, all early region genes and late region structural proteins are encoded, forming a functional BKPyV genome when in consortium. Such consortium may be found in extracellular vesicles in vivo.

Transplant recipients may develop BK polyomavirus-associated nephropathy (BKPyVAN) under diverse clinical conditions, including the presence of decoy cells in urine, the emergence of malignancies in the renourinary tract [32], and, in some cases, rapid loss of graft function in the absence of identifiable causes other than BKPyV infection. Apart from immunosuppression, upon which BKPyV may gain its feared momentum of uncontrolled replication with respect to its reactivation or by primary infection through donor transmission, these heterogeneous clinical manifestations and disease courses appear to be influenced by specific virological properties of the infecting BKPyV strains. These have been shown to differ between compartments—such as urine and blood—even within the same patient [33]. These differences are likely due to viral mutations or genomic rearrangements, some of which have been analyzed earlier in vitro [27].

To investigate the infectivity of virus-laden decoy cell supernatants, we tested samples from multiple patients using the BKPyV-permissive COS WT cell line. Notably, the supernatant from one patient exhibited a particularly high infectivity, transforming COS cells into a stably infected, constitutively virus-producing, and persistently replicating cell line. In contrast, infection with a well-characterized laboratory BKPyV clone might not represent the phenotypic effects observed with clinical decoy cell-derived virus, particularly in terms of secreted viral products and their potential to infect adjacent epithelial cells in the distal urinary tract. These assumptions and previous laboratory experiments [27] highlight the potential for patient-derived BKPyV strains to exert distinct biological effects not captured by standard laboratory clones.

In previous studies, a variety of virustatic agents have been identified using cell-based models for high-throughput screening of large compound libraries and cytotoxicity testing. This approach has proven particularly effective in discovering inhibitors that target viral enzymes. For example, hepatitis C virus (HCV) protease inhibitors were discovered by screening extensive chemical libraries in cell lines inhibiting HCV production, leading to the identification of compounds that inhibit the NS3/4A protease essential for viral replication [34].

In this cell line, an unconventional observation is that the genome harbors an integrated copy of the SV40 large T antigen (LTAg) gene, yet no SV40 virions are produced. Despite the lack of viral particle formation, the integrated SV40 LTAg is actively expressed at the protein level [31]. In this study, we precisely mapped the genomic locus of the integrated SV40 LTAg gene, providing insight into its potential functional consequences.

A notable finding at the protein expression level is that, despite the truncations in the LTAg of one of the BKPyV transcript versions, the LXCXE motif is translated from the ninefold repeated viral genome. This short linear motif is crucial for mediating interactions with the retinoblastoma (Rb) protein, leading to the disruption of cell cycle regulation and the promotion of S-phase entry, thereby facilitating viral replication. This observation aligns with the higher S-phase rate of COSSA cells compared to the parental COS cell line, suggesting a potential mechanistic link between viral integration and altered host cell proliferation dynamics.

When testing for infectivity of the COSSA-secreted virions, the lytic egress phase, characterized by extensive cell damage and subsequent detachment from the culture surface, is observed between the 30th and 34th day in the human tubular cell lines. Although this interval appears long, it likely reflects factors related to the cell model chosen and the specific BKPyV type; however, similar data were shown in another cell line model [20]. This time interval was consistently observed in two independent tubule epithelial cell lines from different patients that had undergone multiple passages in specialized tubule cell culture media.

Various rearrangements in the NCCR have been documented for JCPyV, and these alterations have been linked to the onset of PML [35]. In contrast, the clinical significance of NCCR rearrangements in BKPyV, particularly regarding their role in BKV-associated nephropathy, remain less understood [29,30,36,37]. In the present study, the BKPyV isolate demonstrated efficient replication in a monkey cell line, as well as robust infectivity in human tubular cells, suggesting that NCCR rearrangements may contribute to its infectivity and pathogenic potential, which is in line with work of other research groups.

Several research groups have demonstrated that polyomaviruses can be encapsulated within extracellular vesicles (EVs) [38,39], with each vesicle capable of carrying up to 10 virions [40,41,42]. In this context, it is plausible that distinct, newly rearranged viral genomes may cooperate upon cell entry, facilitating the assembly of functional virions despite the absence of complete structural protein coding in individual genomes. Transcripts derived from one replicon may complement the protein requirements of another genome, thereby enabling successful virion formation and propagation.

This work demonstrates that clinically used diagnostic methods must target more than one locus within the BKPyV minichromosome and must target multiple genetic sites in a highly specific manner; otherwise, some variants, particularly in PyVAN, may be missed. This requirement was met by the most recent authors [43]. If only one antibody is used in diagnostics, such deletions may be important, as the reaction might be negative despite the presence of the viral genome.

In contrast to previously described models, this cell line does not require exogenous virus particles for the infection of established tubular epithelial cell lines [20,44]. Instead, it continuously produces and constitutively secretes BK polyomavirus (BKPyV) particles. This property is attributed to the stable integration of the BKPyV genome into the host cell line. Unlike traditional BKPyV-permissive cell lines—which, while valuable for research, require defined time intervals for productive infection—this model supports high-level, sustained viral replication. Furthermore, the cell line can be maintained in RPMI medium supplemented with 2% fetal calf serum (FCS), which is significantly more cost-effective (by approximately an order of magnitude) than REGM/REBM medium. This is all of relevance in drug screening methods.

It is not well established whether in vivo conditions together with BKPyVAN would reflect exactly these culture conditions. However, any drug that inhibits BKPyV production and egress in this cell line would most likely have an effect and would merit further testing.

A limitation of our experimental approach is the lack of direct characterization of the newly observed non-coding control region (NCCR) rearrangements in the analyzed COSSA cell line. Nevertheless, the potential impact of these rearrangements on viral genome replication remains a compelling area for further investigation. The hypothesis that such alterations may influence viral replication is supported by prior studies demonstrating that specific NCCR rearrangements, particularly those derived from clinical isolates, can significantly modulate BKPyV replication kinetics in experimental systems [27]. Furthermore, due to the absence of single-cell resolution in our study, we infer that the observed viral rearrangements result from the collective representation of the cell population, which likely functions as a cooperative consortium to generate infectious virus particles capable of targeting human cells.

In conclusion, the COSSA cell line constitutes a stable BKPyV-secreting kidney cell line with a well-characterized viral genome integration profile. This model provides a valuable platform for investigating BKPyV biology and may serve as a robust tool for studying the mechanisms and potential inhibition of viral replication.

## 4. Methods

### 4.1. Tissue Culture of Human Proximal Tubular Epithelial Cell Lines and Monkey COS-1 Cell Line

The cells of the urine sediment were cultured in 3.5 cm cell culture dishes using (REBM/REGM) from Lonza (Lonza Group AG, Basel, Switzerland) as culture medium [20]. The BKPyV copy number in the culture supernatant was 10^8^ copies/mL. This supernatant was selected to expose the monkey kidney cell line COS-1 for 3 days. The cell line was maintained in RPMI 1640 with 2% FCS and ampicillin–streptomycin antibiotics. After day 5, many of the cells began to die, and the medium was replaced with fresh culture medium. A small fraction of adherent cells began to proliferate and were passaged after day 12 with trypsin, as described for this cell line. The cell line was then passaged every three days, and virus secretion was monitored from the culture supernatant.

### 4.2. Virus Production Rate in the Persistently BKPyV-Secreting COSSA Cell Line

Reseeding 2 × 10^5^ COSSA cells and supplying them with fresh culture medium leads to a BKPyV shedding of approximately 10^8^ virus units/mL/48 h. These measurements were carried out over at least 10 individual passages, with only slight scatter. The measurements were carried out over 5 months, 7 months, and 12 months after infection. As this quantity of virus production did not change, the establishment and frequency of BKPyV integration into the genome of the cell was targeted as described further down.

### 4.3. DNA Extraction and Purification

Cells from passage 10 were removed from the plates with trypsin and washed with culture media. The cell pellet was then cooled on ice and frozen in a cell freezer at −80 °C on the first day. One day later, the cells were transferred to liquid N_2_. To obtain the DNA, the cells were quickly thawed in a water bath and washed once with PBS. For DNA extraction and purification, the QIAamp^®^ kit (Cat#51104, Hilden, Germany) was employed according to the manufacturer’s protocol. In brief, the lysis/protease reagent provided in the kit was added to the sample, and the resulting lysate was incubated at 56 °C for 10 min in a water bath to ensure complete cell lysis and protein digestion. The pre-cleared lysate was then loaded onto the minicolumn, which was centrifuged at 6000× *g* for 1 min. After thorough washing of the minicolumn, the DNA was eluted using TE buffer and subsequently quantified for concentration and purity with a NanoDrop spectrophotometer.

### 4.4. Immunofluorescence

In addition to monitoring supernatant BKPyV levels, VP1 cellular immunofluorescence was evaluated by a VP1-specific monoclonal antibody (4249, MA5-33242, Invitrogen, Carlsbad, CA, USA). Concomitantly, the SV40-LTAg expression was assessed by an affinity purified anti-SV40 large T-antigen-specific antibody (PA5 112036, Invitrogen, diluted 1:100) (Invitrogen, Waltham, MA, USA). COSSA cells were trypsinated, followed by washing the cell suspension in culture medium. An aliquot of the cell suspension (60 µL) was applied to the funnel of the cytocentrifuge to obtain cytopreparations. After air drying for 60 min, the cytopreparations were fixed in acetone for 5 min. A water repellant circle was drawn with the Super PAP Pen around the cell-containing area. Following a brief wetting of the cells with PBS, the pre-diluted primary antibodies (mouse anti-VP1 mAb 4942, rabbit anti-LTAg, or rabbit anti AQP1 AB2219 Millipore Corp., Burlington, MA, USA) were applied and incubated in a moist chamber at 4 °C overnight. The next morning, the secondary antibodies Alexa-Fluor 488 goat anti-rabbit IgG (A11008, Invitrogen) and Alexa-Fluor 594 goat anti-mouse IgG (H + L) (A11032, Invitrogen) were applied after washing the slides in PBS for 10 min under constant stirring. The secondary antibodies (diluted 1:700 in PBS) containing DAPI for nuclear staining were incubated for 60 min at room temperature. After a final washing procedure with PBS, the slides were mounted with embedding solution and a glass coverslip for image capture by confocal microscopy.

### 4.5. FACS Intracellular VP1 Staining

To investigate the accumulation of intracellular virions and VP1, COSSA cells in suspension were fixed with paraformaldehyde (4%) or 70% ethanol in PBS for at least 1 h. After washing twice in PBS, the primary antibody (mAb 4942 or rabbit anti-BKPyV-VP1 in TPBS, diluted 1:500) was applied and incubated for 90 min at RT on a rolling platform. The cells were washed in TPBS and then incubated with Alexa Fluor goat anti-rabbit 488 or goat anti-mouse Alexa Fluor 594 (diluted 1:400 in PBS) for 60 min with constant rotation. DAPI was added to stain the nucleus. The washed cells were finally suspended in 600 µL PBS and measured with the FACS CANTO.

### 4.6. RNA and Protein Isolation

The culture media were aspirated, and the cell monolayer was washed once with PBS. Cells were lysed using 1000 µL of Trizol. The resulting lysate was kept at room temperature (RT) for at least 10 min, after which 200 µL of chloroform was added. Following mixing by flicking the tube, the lysate was centrifuged at 12,000× *g* for 20 min. The aqueous phase was aspirated for RNA isolation, and 300 µL of ethanol was added and centrifuged at 4000× *g* for 10 min to pellet genomic DNA. The resultant supernatant was mixed with 1500 µL isopropanol and incubated for 15 min at RT. In order to create a protein pellet, the content was centrifuged at 12,000× *g* for 15 min. The protein was then washed three times with 0.3 M guanidine hydrochloride in 95% ethanol. After the final wash, the protein pellet was treated with 95% ethanol and finally air dried. The protein was dissolved in Laemmli sample buffer (BIO-RAD # 1610747) containing 3 M urea for SDS-PAGE gel analysis.

### 4.7. Immunoblotting

Following heating at 95 °C for 3 min, the protein was loaded onto an SDS-PAGE gel and run under reducing condition. Gels were blotted using the Trans-Blot Turbo Transfer System and a setting of constant current 1.8 A, 25 V for 6 min. Following transfer, the membrane was blocked with blocking buffer (PierceTM Protein-Free T20 Blocking Buffer, Appleton, WI, USA, 37571) for 20 min. The blocked membrane was incubated with primary antibody, rabbit anti-VP1 (dil:1:1500 in blocking buffer) [45], at 4 °C overnight. The next morning, the membrane was washed twice for 10 min in TPBS and incubated in the HRP-conjugated goat anti-rabbit Ab (dil. 1:10,000). The antibody diluent consisted of PBS/PierceTM Protein-Free T20 Blocking Buffer at a ratio of 9:1. After two final washes in TPBS, the antibody binding was visualized using a chemiluminescence reagent (Merck KGaA, Darmstadt, Germany) and recorded by lumi-imaging with Fusion FX Vilber Lourmat (Vilber, Eberhardzell, Germany).

### 4.8. Testing the Infectivity of COSSA-Secreted Virus Particles

Three tubular progenitor cell lines, as described earlier, were cultured in 3.5 cm culture dishes using proximal tubular cell culture media (REBM/REGM) obtained from Lonza (Lonza Group AG, Basel, Switzerland) [20,24]. These cell lines were exposed to 1000 µL of COSSA cell supernatant containing 1 × 10^8^ copies/mL. Cell morphology and copy number in the tissue culture supernatant were monitored until day 34 post-infection.

### 4.9. BKPyV Copy Number Evaluation

Sample processing was performed with the GeneProof PathogenFree DNA Isolation Kit (Vídeňská 101/119/Dolní Heršpice/CZ-619 00 Brno).

The BK/JC test was performed as described in the company’s (GeenProof, GeneProof BK/JC Virus (BK/JC)) test kit manual. In brief: Master Mix (30 µL) was dispensed into each PCR tube; subsequently, 10 μL of the extracted test sample nucleic acid sample or 10 μL of the calibrator was added to the respective PCR tubes. After thorough mixing by pipetting to ensure homogeneity, the final reaction volume was 40 μL. This was placed into the real-time PCR thermocycler and amplified using the specified PCR cycling conditions. The formula to calculate the viral load concentration relying on the PCR amplification values in IU/mL is provided in the test manual (www.geneproof.com).

### 4.10. ONT Sequencing and Bioinformatic Analysis

The extracted DNA sample with a total of ~6.8 µg DNA was left side size selected to >10 kb using the Circulomics Short Read Eliminator Kit XS. In brief: The volume of DNA sample was mixed with an equal volume of Circulomics XS buffer and incubated at 56 °C for 1 h. The sample was centrifuged at 10,000× *g* for 30 min at room temperature. The pellet was washed twice with 70% EtOH and centrifuged at 10,000× *g* for 2 min at room temperature, and the supernatant was discarded. The sample was incubated with ddH_2_O for 20 min at room temperature to redissolve the DNA pellet. After left side size selection, 3.8 µg of DNA remained. The entire DNA sample was then used for library preparation using the SQK-LSK114 Library prep kit (Oxford Nanopore Technologies), and a whole-genome sequencing run was performed with a PromethION P2-Solo on a Promethion flow cell.

The basecalling was carried out with the Dorado basecaller (v0.7.1) for super accuracy. The subsequent mapping was carried out with Minimap2 (v2.28-r1209) and Samtools (v1.16.1). NCBI’s BLAST tool (v2.14.0) was employed for sequence similarity searches. The reads were mapped against the monkey-reference: Chlorocebus aethiops (green monkey, NCBI taxonomy: Genome assembly ASM2378351v1, https://www.ncbi.nlm.nih.gov/datasets/genome/GCA_023783515.1/, accessed on 25 November 2024). To assess potential bias from endogenous viral elements (EVEs), the viral sequences integrated into the host genome and transmitted vertically, the genome assembly was screened for EVEs using detectEVE v1.4.1. Subsequently, all mapped reads were blasted against the virus reference: BK polyomavirus DNA (complete genome, isolate: CAF-15, NCBI genebank: AB263912.1). A de novo assembly with Flye (v2.9.1-b1780) and Medaka polishing (v1.7.2) was performed over the positive blast results, and the resulting contigs were analyzed for the integration sites of the virus into the monkey genome. Subsequently, the reads were also tested for the integration of Simian virus 40 (SV40, strain 776, NCBI gene bank AF316139.1) by blast analysis, de novo assembly, and mapping.

An average coverage of 30× was reached over the monkey genome, the N50 of the whole sequencing run was 30 kb, and the average Phred-score was 24. From 144 gigabases, 85 gigabases were aligned against the monkey-reference. The reads that did not align to the monkey-reference had an N50 of 3476 bp. No EVEs were found in the reference, ensuring no bias from pre-existing viral sequences.

### 4.11. Bioinformatical Workflow for Viral Genome Construct

The sequenced DNA was basecalled using the previously established protocol and subsequently aligned against the reference viral genome (AB263912.1; BK polyomavirus DNA, complete genome, isolate CAF-15) using Minimap2 v2.28 with the following parameters: (minimap2-ax lr:hq-Y). Softclips larger than 200 bp were extracted (using jvarkit samextractclip) from the alignment data and aligned against the host genome using minimap2 as before, all reads aligning to the monkey genome were removed from the first dataset, and a histogram of the sequence length was produced utilizing an in-house python script using numpy and matplotlib for analysis and plotting. Based on the peaks observed in the histogram, sequences ranging from 4530 to 4550 and 4810 to 4830 bp were extracted, and a de novo assembly was generated using Flye. The assembly was performed using the following parameters (flye–nano-hq input-o outpu -g 5 k–asm-coverage 500–m 4000). This approach ensured a high-fidelity reconstruction of the circularized BKPyV genomes while maintaining a depth of coverage suitable for the detection of structural variants.

To identify the BKPyV clinical isolate used for COS-WT infection, freshly extracted viral DNA was subjected to separate sequencing. Reads ranging in length from 5040 to 5240 base pairs were selected for de novo assembly to reconstruct the viral genome. For this, Flye was used for assembly with the following parameters: flye --nano-hq input -o output -g 5 k --asm-coverage 500 -m 4000. This approach enabled high-fidelity reconstruction of the circular BK polyomavirus (BKPyV) genome. The assembled circular BKPyV genome had a total length of 5141 bp, supported by a sequencing coverage depth of 12,788 for the NCCR and 5000 for the other parts. The IGV pile-up is shown in Appendix A.

## Figures and Tables

**Figure 1 ijms-26-05745-f001:**
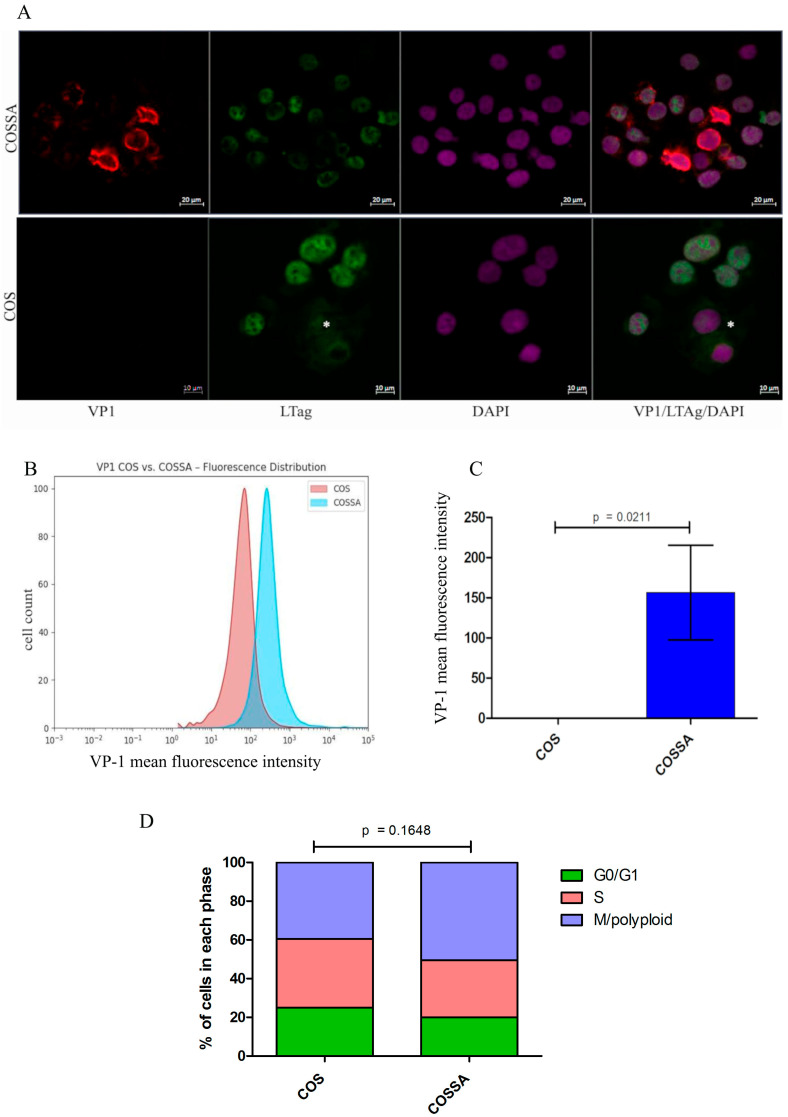
(**A**) Immunofluorescence analysis of COSSA and COS-WT cells for BKPyV VP1 and LTAg expression. Cytopreparations of COSSA cells (upper panel) and parental COS-WT cells (lower panel) were stained with monoclonal antibody 4942 targeting the BKPyV capsid protein VP1 (red) and a rabbit anti-SV40 antibody recognizing large T antigen (LTAg, green) following acetone fixation. Cells with high levels of VP1 (red) are indicative of extensive virion production. A cell in telophase displays cytoplasmic redistribution of LTAg, as indicated by an asterisk (*****). (**B**) Intracellular VP1 staining after fixation and permeabilization of the cells and FACS measurement. COS and COSSA cells were fixed in 70% ethanol and stained with monoclonal antibody 4942, followed by a fluorophore-conjugated goat anti-mouse secondary antibody. BKPyV-infected COSSA cells (blue histogram) showed a significant increase in VP1 fluorescence intensity compared to the parental COS cell line (red histogram). (**C**) Statistical analysis was performed using the Mann–Whitney U test (*p* = 0.0211, n = 4). To adjust for non-specific background fluorescence, the mean background fluorescence intensity measured in COS cells was subtracted from the values obtained for both COS and COSSA cells in (**C**). (**D**) Cell cycle analysis by intracellular FACS of parental COS-WT cell line and the infected COSSA cell line. COS-WT and COSSA cells were fixed with paraformaldehyde, permeabilized with TPBS, stained with DAPI, and analyzed by flow cytometry. Cell cycle analysis comparing COS and COSSA cells demonstrated an increased proportion of COSSA cells in the mitotic/polyploid phase 24 h after passaging. However, this difference was not statistically significant (two-tailed unpaired Student’s *t*-test, n = 6, *p* = 0.1648).

**Figure 2 ijms-26-05745-f002:**
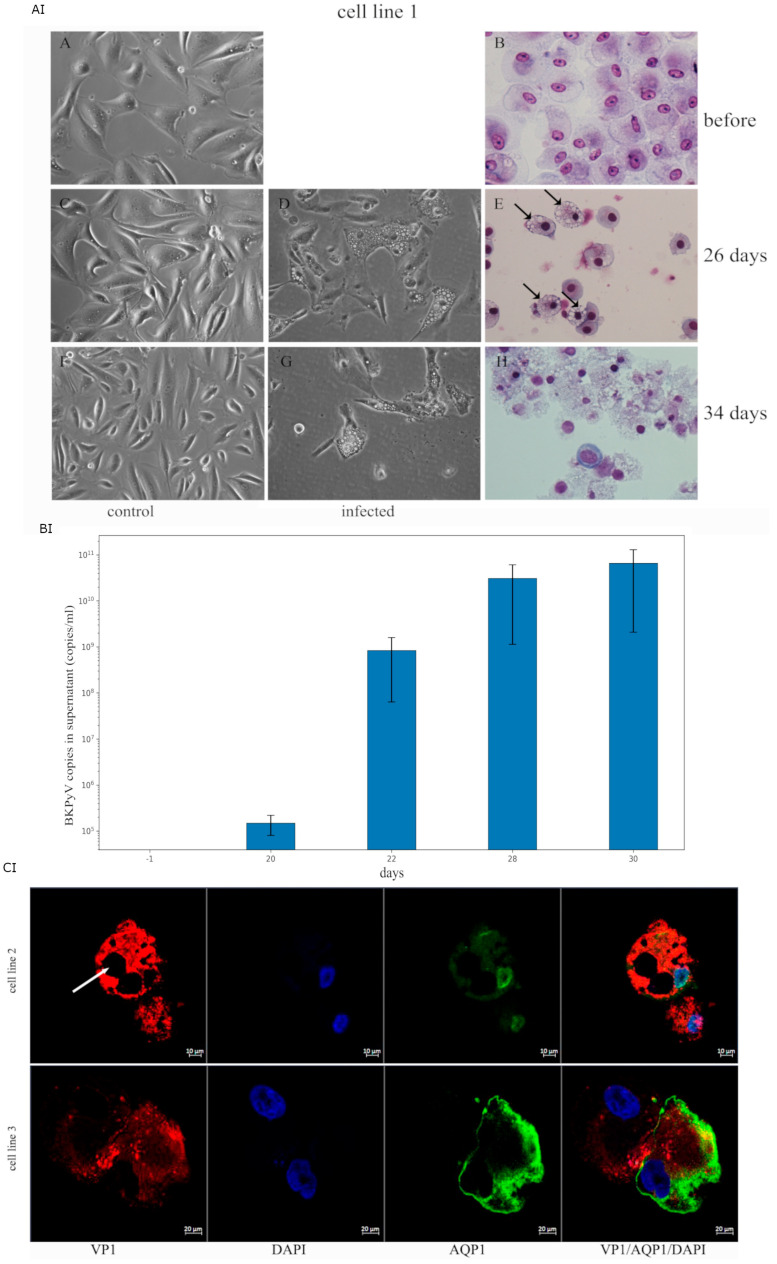
(**AI**) Phase-contrast and hematoxylin and eosin (H,E) staining of a renal tubular epithelial progenitor cell line #1 infected with COSSA supernatant and control. Under culture conditions before infection (A,B) and control (C), no vacuolation was observed. BKPyV-exposed tubular progenitor cell line cells exhibited progressive morphological alterations characterized by the accumulation of intracellular lipoid vacuoles which became more prominent by day 20–26, as demonstrated by phase-contrast microscopy (D) and hematoxylin and eosin (H,E) staining (E). Arrows in panel (E) highlight the vacuolar structures. By day 34, no vacuolation was observed under control culture conditions (F), whereas in the experimental infection, vacuolation was more prominent ((G) phase contrast)) and was accompanied by cell lysis and detachment from the culture dish. Detached cells showed marked degenerative changes, as confirmed by H,E staining (H). (**BI**) Course of BKPyV copies in tissue culture supernatant after tubular progenitor cell line #1 infection with COSSA-derived BKPyV. The virus causes dramatic changes in the human renal epithelial cell line, which starts at days 5–7 post-infection and reaches its maximum at days 26–34 when the cells start to detach. These changes observed in phase contrast microscopy coincide with increased virus particle release, increasing at day 26 to 10^9^ units/mL and reaching a peak of 10^11^ at day 34, when more than 95% of the cells were lysed and detached and positive by VP1 staining. Each time point represents the mean ± standard deviation (SD) from two independent experiments using different cell lines. At day −1, both the cell lines and their culture supernatants tested negative for BKPyV. (**CI**) Dual-color confocal microscopy assessing intracellular BKPyV VP1 expression in combination with aquaporin-1 (AQP1) in tubular progenitor cell lines #2 and #3 infected with COSSA-derived BKPyV. Following acetone fixation, cells were immunostained using the BKPyV VP1-specific monoclonal antibody mAb 4942 (red) and a rabbit polyclonal anti-AQP1 antibody (green). Strong intracellular VP1 staining was observed in both tubular progenitor cell lines (upper and lower panels). Marked vacuolization, as indicated by the arrow, is shown in the upper panel (cell line #2). Nuclear staining was performed with DAPI (blue). Corresponding negative controls for VP1 staining from both cell lines are presented in the Appendix A. Prior to this infection experiment, the cell line #2 shown in the upper panel was confirmed to be BKPyV-negative by sequencing to exclude bias through endogenous reactivation.

**Figure 3 ijms-26-05745-f003:**
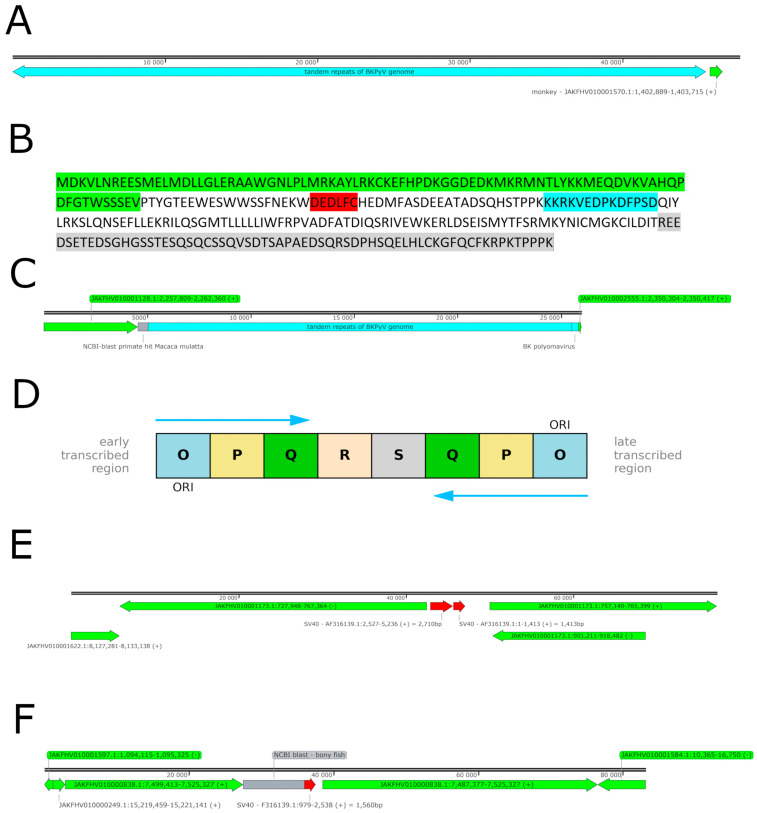
(**A**) Integration of BKPyV genome in tandem repeats. Nine tandem repeats of the BKPyV genome spanning over 45,481 bp (blue) were identified, each lacking a fragment of the LTAg coding region. Notably, this deletion is consistent with the contigs obtained from the sequenced viral genomes destined for particle formation; see below. The flanking monkey genome is depicted in green. (**B**) Truncated LTAg protein. The primary structure of the BKPyV-encoded LTAg protein exhibits a 1224 bp deletion at the nucleotide level in contig 4; however, the reading frame remains intact when spliced together. Consequently, the J-domain (green), along with the LXCXE motive (red), a 10-amino-acid-long part of the origin binding domain (OBD) (blue), a short part of the helicase, and the host interaction domain, which binds to p53 (gray), are preserving their structural integrity and alignment with the canonical LTAg protein. (**C**) Integrated BKPyV genome in tandem repeats. Bioinformatic analysis of a contiguous 25,964 bp sequence revealed the integration of four tandem repeats of the BKPyV genome (blue), with the fourth repeat exhibiting truncation. Flanking sequences corresponding to the monkey host genome (green) were also identified, confirming the integration event. The fragment shown in gray could not be unequivocally assigned to a specific genomic region. (**D**) Rearranged NCCR in COSSA. A schematic representation of the block structure of the BKPyV-NCCR as observed in the fourfold tandem repeat. Notably, the rr-NCCR exhibits a novel rearrangement, with the first 238 bp forming an inversion of regions O, P, and Q following the S segment relative to the canonical sequence (indicated by reversed arrow). This structural alteration may have significant functional implications for the regulation of viral replication. (**E**) Integration of SV40 genomic fragments into the monkey genome. The integrated SV40 fragment, highlighted in red against the monkey genome (shown in green), spans 4124 bp (contig A, Table 1). This segment encompasses the complete coding sequence for the large T antigen (LTAg) along with its promoter, the agnoprotein coding region, and partial sequences corresponding to VP2 and VP3. (**F**) Integration of SV40 genomic fragments into the monkey genome. The 1560 bp SV40 fragment (red) is shown in the genetic environment of the monkey genome (green). It contains a coding part of the VP1, VP2, and VP3 proteins, but does not provide any promoter for transcription. The portion depicted in gray denotes a sequence that could not well be aligned (contig B, Table 1).

**Figure 4 ijms-26-05745-f004:**
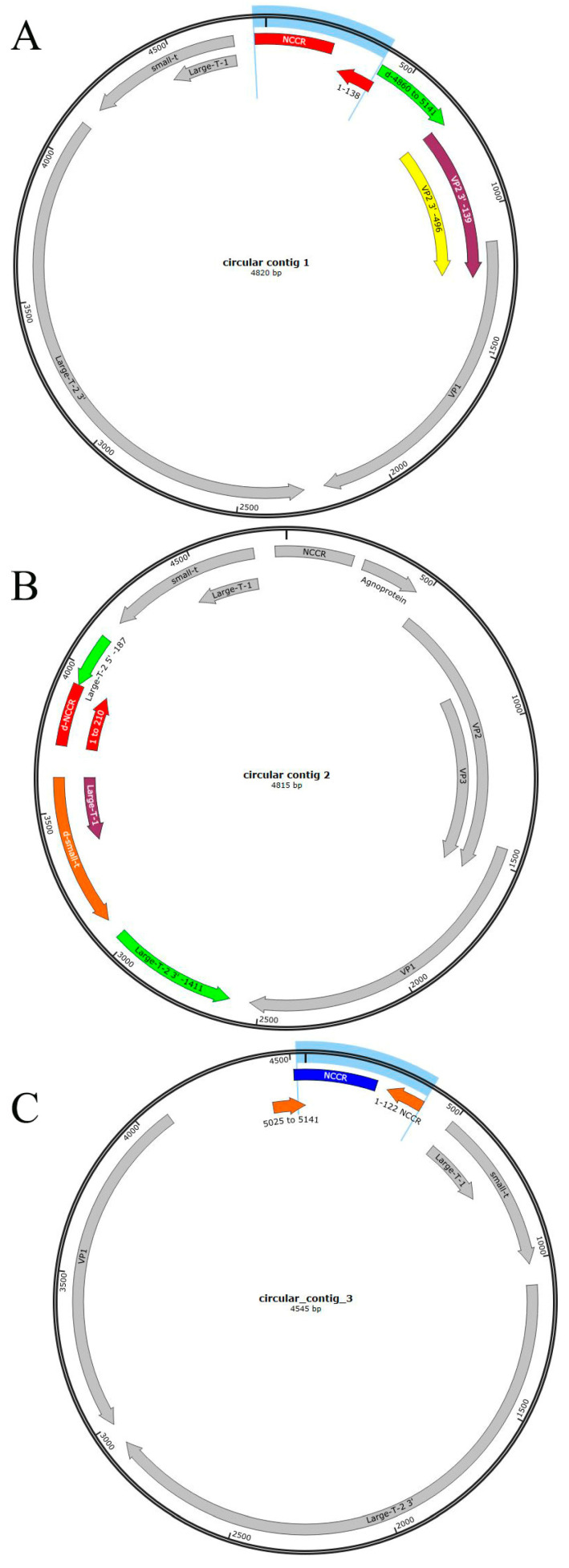
(**A**) Circular BKPyV genome contig of 4820 bp. Structural arrangement of BKPyV circular genome of 4820 bp featuring an rr-NCCR rearrangement and lacking agnoprotein coding. The rr-NCCR exhibits an inversion of the first 138 bp (marked in red) of the original NCCR at its 5′ end, leading to a duplication of the origin of replication (ORI), replication enhancers, and transcription factor binding sites. The extended rearranged-NCCR (red, rr-NCCR) is shown with blue brackets. Translocation of bp 4860-5141 is marked in green. VP2 (yellow) and VP3 (purple) are truncated. (**B**) Circular BKPyV genome contig of 4815 bp. The BKPyV circular genome of 4815 bp contains an NCCR repeat and retains the coding sequence for the agnoprotein along with the structural proteins VP1, VP2, and VP3, while exhibiting a disrupted LTAg coding sequence. The LTAg coding region is interspersed with this NCCR repeat fragment and an sTAg duplicate. This genomic variant originates from a replication site at the nonameric tandem repeat, spanning nucleotides 39,673–44,487. The alterations in the coding are marked in color. (**C**) Circular BKPyV genome contig of 4545 bp. BKPyV circular genome of 4545 bp featuring an rr-NCCR rearrangement similar to that of circular contig 1 and lacking agnoprotein, VP2, and VP3 coding. The rr-NCCR exhibits an inversion of the first 122 bp of the original NCCR at its 5′ end, leading to a duplication of the origin of replication (ORI). Structural variation/rearrangement is shown in coloration. The coding of late and early genes is reversed. The extended rr-NCCR is shown with blue brackets.

**Table 1 ijms-26-05745-t001:** Distribution of host genomic DNA versus integrated polyomavirus DNA and supporting read coverage identified by ONT. Total length of reads, proportion of PyV means: and read fragment length of polyomavirus-specific DNA within the host sequence. Coverage of reads: number of reads covering each contig. The asterisk (*) indicates this contig was found in two further workflows, four times in each.

Contig	Total Length	Proportion PyV	Coverage of Reads	Type
1 *	19,291	1200	4	BKPyV
2	14,584	2944	20	BKPyV
3	4155	1668	20	BKPyV
4	47,651	45,481	22	BKPyV
5	25,964	20,460	109	BKPyV
A	77,115	4124	25	SV40
B	83,050	1560	39	SV40

## Data Availability

All data are provided within this article.

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
