# Peer review of "Establishment of a Stable BK Polyomavirus-Secreting Cell Line: Characterization of Viral Genome Integration and Replication Dynamics Through Comprehensive Analysis"

_ijms, 2025, doi:10.3390/ijms26125745_

Round 1

Reviewer 1 Report

Comments and Suggestions for Authors

This study describes a method to establish a BKV-secreting cell line and its potential applications. While the study is technically rigorous and addresses a critical gap in polyomavirus research, several issues require clarification to strengthen the conclusions and ensure reproducibility.

  1. The Title: The title emphasizes the application of the system for antiviral drug screening. However, this study didn’t include any antiviral data. So the authors should either modify the title or add some antiviral data
  2. Viral Genome Heterogeneity
  • The study identifies multiple BKPyV variants with structural rearrangements (e.g., NCCR inversions) in COSSA cells. The authors should detail whether the COSSA was established from a single viral clone or a mixed viral population. This is essential to determine whether the observed rearrangements originated from the clinical isolate or occurred during cell line adaptation.
  • The authors should discuss why a clinical isolate was chosen directly rather than a well-characterized typical strain? What is the rationale or advantage of using a patient-derived strain for establishing COSSA? Clarifying this will help readers assess the relevance and reproducibility of the model.
  1. Data Presentation, Figures, and Statistical Rigor

The overall figure layout and content require reorganization to enhance logical flow and highlight key findings. Please revise:

  • All figures must include essential elements such as informative titles, axis labels (with units, e.g., for qPCR in Figure 1), legends, and scale bars where applicable.
  • Quantitative experiments (e.g., FACS, qPCR) lack detailed statistical reporting. Please indicate the number of biological and technical replicates, display error bars, and provide p-values with corresponding statistical tests (e.g., Figures 3B and 4).
  1. Misleading or Unsupported Statements
  • Lines 56–58: The statement “With the exception of the large tumor antigen (LTAg) and its structural proteins…” is inaccurate, as all viral protein synthesis—including LTAg and structural proteins—requires host machinery. Please revise this to reflect accurate virology principles.
  • Line 265: The claim that LTAg expression is “downregulated due to virion formation” is not sufficiently supported. LTAg is typically expressed early and then declines as replication progresses. Either provide supporting literature or revise the statement to reflect known viral replication dynamics.
  • The hypothesis that NCCR rearrangement enhances BKPyV replication is plausible but remains speculative in the current study, as no functional validation experiments were performed. It is recommended that the authors explicitly discuss this limitation in the Discussion section, referencing relevant literature where similar rearrangements have been linked to increased viral replication or pathogenicity. Such contextualization will help ground the hypothesis and acknowledge the need for future experimental validation (e.g., reporter assays or replicative fitness comparisons between wild-type and rearranged NCCR constructs).

  1. Methodological Clarity – ONT Sequencing and COSSA Establishment
  • Section 2.9 (Nanopore Sequencing): Critical information such as sequencing depth, integration call thresholds, and mapping criteria is missing. Please provide sufficient methodological details to ensure reproducibility.
  • Lines 229–235: The description of COSSA establishment currently appears in the Results section, but it is methodological in nature. This content should be moved to the Methods section (likely under “2.1 Cell Culture”) to maintain structural clarity.

  1. Reference Format

References are inconsistently formatted. Please standardize citation style throughout the manuscript in accordance with journal guidelines.

Recommendation

This study presents a technically advanced and clinically relevant model for BKPyV research. However, addressing the major concerns (e.g., functional validation of integration sites, statistical rigor) and minor revisions (e.g., figure clarity) are essential before publication. With these revisions, the manuscript will make a significant contribution to the field.

Author Response

We would like to thank the reviewer for the positive and constructive comments. We have carefully incorporated the reviewer’s suggestions, which have significantly enhanced the clarity and overall quality of the manuscript.

This study describes a method to establish a BKV-secreting cell line and its potential applications. While the study is technically rigorous and addresses a critical gap in polyomavirus research, several issues require clarification to strengthen the conclusions and ensure reproducibility.

  1. The Title: The title emphasizes the application of the system for antiviral drug screening. However, this study didn’t include any antiviral data. So the authors should either modify the title or add some antiviral data

We have modified the title accordingly.

  1. Viral Genome Heterogeneity
  • The study identifies multiple BKPyV variants with structural rearrangements (e.g., NCCR inversions) in COSSA cells. The authors should detail whether the COSSA was established from a single viral clone or a mixed viral population. This is essential to determine whether the observed rearrangements originated from the clinical isolate or occurred during cell line adaptation.

This suggestion is well taken and data in this respect were newly generated and are included in this version of the manuscript. We have managed to sequence the original virus, which has been used for COS-WT infection in subsection 3.7.

To determine whether the COSSA cell line originated from a single viral clone or a heterogeneous viral population, viral DNA from the inoculum was extracted and subjected to Oxford Nanopore Technologies (ONT) long-read sequencing. The analysis identified a canonical BKPyV genome of 5,141 base pairs, harboring nucleotide substitutions at four distinct positions, indicating the presence of a minor mutated variant within the initial viral population. No genomic rearrangements were detected. The full BKPyV sequence and the identified nucleotide variants are presented in the supplementary data, along with the corresponding IGV pile-up visualization.

  • The authors should discuss why a clinical isolate was chosen directly rather than a well-characterized typical strain? What is the rationale or advantage of using a patient-derived strain for establishing COSSA? Clarifying this will help readers assess the relevance and reproducibility of the model.

In the discussion, we elaborate on the rationale and objectives underlying our decision to utilize patient-derived samples, emphasizing their relevance for capturing the biological diversity and clinical complexity of BKPyV infection that may not be adequately represented by laboratory-adapted viral clones. These isolates reflect the capacity to generate decoy cells in a subset of patients with BKPyV-associated nephropathy (BKPyVAN).

Transplant recipients may develop BK polyomavirus-associated nephropathy (BKPyVAN) under diverse clinical conditions, including the presence of decoy cells in urine, the emergence of malignancies in the renourinary tract, and, in some cases, rapid loss of graft function in the absence of identifiable causes other than BKPyV infection. These heterogeneous clinical manifestations and disease courses appear to be influenced by specific virological properties of the infecting BKPyV strains, which have been shown to differ between compartments—such as urine and blood—even within the same patient [1]. These differences are likely due to viral mutations or genomic rearrangements of which some have been analyzed earlier [2]. To investigate the infectivity of virus-laden decoy cell supernatants, we tested samples from multiple patients using the BKPyV-permissive COS cell line. Notably, the supernatant from one patient exhibited a particularly high infectivity, transforming COS cells into a stably infected, constitutively virus-producing, and persistently replicating cell line. In contrast, infection with a well-characterized laboratory BKPyV clone might not replicate the phenotypic effects observed with clinical decoy cell-derived viruses, particularly in terms of secreted viral products and their potential to infect adjacent epithelial cells in the distal urinary tract. These findings highlight the potential for patient-derived BKPyV strains to exert distinct biological effects not captured by standard laboratory clones.

Data Presentation, Figures, and Statistical Rigor

The overall figure layout and content require reorganization to enhance logical flow and highlight key findings. Please revise:

  • All figures must include essential elements such as informative titles, axis labels (with units, e.g., for qPCR in Figure 1), legends, and scale bars where applicable. Quantitative experiments (e.g., FACS, qPCR) lack detailed statistical reporting. Please indicate the number of biological and technical replicates, display error bars, and provide p-values with corresponding statistical tests (e.g., Figures 3B and 4).

All figures presented include titles in the legend head line, statistical analyses and are based on experiments performed in triplicate or quadruplicate to ensure reproducibility and reliability of the data. Number of biological replicates are indicated in the figure legend.

  1. Misleading or Unsupported Statements
  • Lines 56–58: The statement “With the exception of the large tumor antigen (LTAg) and its structural proteins…” is inaccurate, as all viral protein synthesis—including LTAg and structural proteins—requires host machinery. Please revise this to reflect accurate virology principles.

We thank the reviewer for indicating this missleading staement. We have corrected the wording.

  • Line 265: The claim that LTAg expression is “downregulated due to virion formation” is not sufficiently supported. LTAg is typically expressed early and then declines as replication progresses. Either provide supporting literature or revise the statement to reflect known viral replication dynamics.

We agree with the reviewer’s comment and have revised the descriptive statement to reflect that cells exhibiting high VP1 expression show reduced LTag staining intensity. No further conclusion is taken from this.

The hypothesis that NCCR rearrangement enhances BKPyV replication is plausible but remains speculative in the current study, as no functional validation experiments were performed. It is recommended that the authors explicitly discuss this limitation in the Discussion section, referencing relevant literature where similar rearrangements have been linked to increased viral replication or pathogenicity. Such contextualization will help ground the hypothesis and acknowledge the need for future experimental validation (e.g., reporter assays or replicative fitness comparisons between wild-type and rearranged NCCR constructs).

We thank the reviewer for this valuable comment and have addressed this limitation of our study in the discussion section. To support our hypothesis, we also cite prior work by established investigators demonstrating relevant findings that align with our observations.

.

A limitation of our experimental approach is the lack of direct characterization of the newly observed non-coding control region (NCCR) rearrangements in the analyzed COSSA cell line. Nevertheless, the potential impact of these rearrangements on viral genome replication remains a compelling area for further investigation. The hypothesis that such alterations may influence viral replication is supported by prior studies demonstrating that specific NCCR rearrangements, particularly those derived from clinical isolates, can significantly modulate BKPyV replication kinetics in experimental systems [2].

  1. Methodological Clarity – ONT Sequencing and COSSA Establishment
  • Section 2.9 (Nanopore Sequencing): Critical information such as sequencing depth, integration call thresholds, and mapping criteria is missing. Please provide sufficient methodological details to ensure reproducibility.

Further information is provided in the methods section on ONT sequencing.

An average coverage of 30x was reached over the monkey-genome, the N50 of the whole sequencing run was 30kb and the average phred-score was 24. From 144 Gigabases 85 gigabases were aligned against the monkey-reference. The reads which didn’t align to the monkey-reference had a N50 of 3,476 bp. No EVEs were found in the reference, ensuring no bias from pre-existing viral sequences.

  • Lines 229–235: The description of COSSA establishment currently appears in the Results section, but it is methodological in nature. This content should be moved to the Methods section (likely under “2.1 Cell Culture”) to maintain structural clarity.

We have integrated this paragraph into the methods section

  1. Reference Format

References are inconsistently formatted. Please standardize citation style throughout the manuscript in accordance with journal guidelines.

The inconsistencies in the reference citation format have been corrected and updated to MDPI mode.

References

  1. Mineeva-Sangwo, O.; Van Loon, E.; Andrei, G.; Kuypers, D.; Naesens, M.; Snoeck, R. Time-dependent variations in BK polyomavirus genome from kidney transplant recipients with persistent viremia. Sci Rep 2023, 13, 13534, doi:10.1038/s41598-023-40714-4.
  2. Broekema, N.M.; Abend, J.R.; Bennett, S.M.; Butel, J.S.; Vanchiere, J.A.; Imperiale, M.J. A system for the analysis of BKV non-coding control regions: Application to clinical isolates from an HIV/AIDS patient. Virology 2010, 407, 368-373, doi:https://doi.org/10.1016/j.virol.2010.08.032.

Reviewer 2 Report

Comments and Suggestions for Authors

Although I find the results very interesting, the article lacks quality in the presentation of the data and requires the inclusion of additional controls. The changes I suggest are:

  1. The introduction needs to be improved so that readers understand why the authors chose the COS-1 cell line over others (also explain that COS was transformed with SV40 LT, this in order to understand results later) , the importance of renal proximal tubular epithelial cells, and what other cells have been used to study BKPyV. Recommended Reading:   Front Endocrinol (Lausanne) . 2022 Apr 8:13:834187. doi: 10.3389/fendo.2022.834187.PLoS Pathog . 2019 Jan 8;15(1):e1007505. doi: 10.1371/journal.ppat.1007505.J Med Virol. 2024 Nov;96(11):e70038. doi: 10.1002/jmv.70038.
  2. It needs to be clear in all figures that at least three independent experiments were carried out. Sometimes, the significance of the findings is not clear (un least not required), Sometimes I was not sure about the significance of the findings

  3. The content in Figures 1 and 2 is not relevant to this article. However, the author could consider using this material as supplementary content and discussing it in the discussion section.
  4. Figure 3 lacks negative controls and the display of single channels. Additionally, the scientific explanation for the reasons behind the measurements is not clear to me, I think it will be better to just present the result and maybe use them for discussion.  (differences between LT and VP1 expression correspond to features of life cycle (early and late genes; more over  you need to clarify that using this method you can not distinguish between expression of LT from SV40  -which is inserted also in COS-1 genome or the BK)
  5. Overall, the figures have very low resolution, and some of the descriptions are not precise. I recommend improving the graphical display and figure legends.
  6. The authors did not show the virus, so it is still possible that only VP1 is being expressed but does not encapsidate defective genomes (e.g., long genomes). This could result in particles with VP1 but no genomes. It is necessary to carry out electron microscopy on the supernatants to confirm the presence of the full virus (not just empty particles).

  7. When using proximal tubular epithelial cells isolated from patients, it is essential to show the negative controls at every time point. There is a possibility that endogenous bk virus within the cells is being multiplied. Therefore, negative controls should be included in Figure 5 at every time point to rule out this possibility

  8. Finally, I believe you cannot exclude the possibility that episomal DNA is still contributing to the infection
Comments on the Quality of English Language

As I am not a native speaker, I recommend using proofreading services to ensure the final version is polished and accurate

Author Response

We would like to thank the reviewer for the constructive comments and valuable criticisms. In response, we have revised the current version of the manuscript by improving both the figures and the main text. All changes are highlighted in yellow for clarity. As suggested, a subset of the original figures has been moved to the supplementary material and, where appropriate, replaced by newly generated figures as requested. Our detailed responses to each of the reviewer’s comments are provided in a point-by-point format below.

Although I find the results very interesting, the article lacks quality in the presentation of the data and requires the inclusion of additional controls. The changes I suggest are:

  1. The introduction needs to be improved so that readers understand why the authors chose the COS-1 cell line over others (also explain that COS was transformed with SV40 LT, this in order to understand results later) , the importance of renal proximal tubular epithelial cells, and what other cells have been used to study BKPyV.

Several revisions were made to the introduction, including a detailed description of the origin, development, and research utility of COS cell lines.

A critical development was the generation of COS cell lines, achieved by transfecting the parental African green monkey kidney CV-1 cell line with an origin-deficient SV40 ge-nome. These modified cell lines are incapable of producing infectious SV40 particles [1,2]. However, subsequent studies revealed that COS cells remain permissive to replica-tion-defective SV40 mutants, thus enabling their replication and maintenance within the cells and thereby supporting their propagation.

  1.  Recommended Reading:   Front Endocrinol (Lausanne) . 2022 Apr 8:13:834187. doi: 10.3389/fendo.2022.834187.PLoS Pathog . 2019 Jan 8;15(1):e1007505. doi: 10.1371/journal.ppat.1007505.J Med Virol. 2024 Nov;96(11):e70038. doi: 10.1002/jmv.70038. Thank you for recommending these readings. We have followed up on them and included additional data derived from our own investigations using tubular epithelial progenitor cell lines in the results section. To avoid any potential misunderstandings, we would like to clarify that we did not employ primary tubular epithelial cells. Instead, we used established tubular progenitor cell lines derived from single-cell outgrowths. The generation and characterization of these cell lines have been described in detail in our previous publications. All of them were BKPyV negative one of them even by ONT genome sequencing.
  2. It needs to be clear in all figures that at least three independent experiments were carried out. Sometimes, the significance of the findings is not clear (un least not required), Sometimes I was not sure about the significance of the findings Thank you for the suggestion. We have revised the figure accordingly and added a statement in the text indicating that at least three independent experiments were performed, with statistical analysis described in the corresponding section.
  3. The content in Figures 1 and 2 is not relevant to this article. However, the author could consider using this material as supplementary content and discussing it in the discussion section. We concur with the reviewer’s comment and have accordingly moved the respective figures to the supplementary section.
  4. Figure 3 lacks negative controls and the display of single channels. Additionally, the scientific explanation for the reasons behind the measurements is not clear to me, I think it will be better to just present the result and maybe use them for discussion.  (differences between LT and VP1 expression correspond to features of life cycle (early and late genes; more over  you need to clarify that using this method you can not distinguish between expression of LT from SV40  -which is inserted also in COS-1 genome or the BK)

We thank the reviewer for this helpful comment. In response, we have revised the figure to display all individual channels separately and have included a negative control using the parental COS cell line. Corresponding statements in the Results section have been updated and modified as suggested by the reviewer accordingly. These are highlighted in yellow.

  1. Overall, the figures have very low resolution, and some of the descriptions are not precise. I recommend improving the graphical display and figure legends.

We have revised the figures and placed greater emphasis on providing detailed and accurate descriptions in the figure legends to improve clarity and interpretability.

  1. The authors did not show the virus, so it is still possible that only VP1 is being expressed but does not encapsidate defective genomes (e.g., long genomes). This could result in particles with VP1 but no genomes. It is necessary to carry out electron microscopy on the supernatants to confirm the presence of the full virus (not just empty particles).

Due to the unavailability of electron microscopy (ELMI), we characterized the isolated particles using three different BK polyomavirus (BKPyV)-specific probe sets targeting the Agno, VP1, and VP2 structural proteins, as well as the Large T antigen (LTAg), through both quantitative and conventional PCR. All probe sets yielded strongly positive results, confirming the presence of BKPyV sequences. Additionally, BKPyV DNA was quantified in the tissue culture supernatant using the GENE Proof assay—a clinically validated method routinely employed for the detection of BKPyV DNA in patient urine and blood samples. The reviewers’ in-depth analysis and expertise are greatly appreciated. However, our methodological data support the conclusion that the particles in question are unlikely to be devoid of the full BKPyV genome. Furthermore, we demonstrate that particles derived from COSSA cells are capable of infecting renal tubular epithelial cell lines that had previously tested negative for BKPyV. Notably, one of these cell lines had undergone prior sequencing and was confirmed to be free of BKPyV genomic material. The others were immunologically negative.

  1. When using proximal tubular epithelial cells isolated from patients, it is essential to show the negative controls at every time point. There is a possibility that endogenous bk virus within the cells is being multiplied. Therefore, negative controls should be included in Figure 5 at every time point to rule out this possibility

In the revised version of the manuscript, we clearly specify that tubular progenitor cell lines—not primary tubular epithelial cells—were used in our experiments. In Figure 4, we now include a control cultured under identical conditions and time points, but without prior BKPyV infection, to serve as a direct comparison and control.

  1. Finally, I believe you cannot exclude the possibility that episomal DNA is still contributing to the infection.

We thank the reviewer for carefully reading our manuscript. We believe we present a strong argument against the possibility that BKPyV activation resulted merely from pre-existing episomal viral DNA during the observation period. Specifically, whole-genome sequencing of the cell line used which was performed before starting the experiment confirmed the absence of any BKPyV genomic sequences or viral gene fragments. Furthermore, no immunologic evidence of virus could be detected in control conditions.

References

  1. Gluzman, Y. SV40-transformed simian cells support the replication of early SV40 mutants. Cell 1981, 23, 175-182, doi:10.1016/0092-8674(81)90282-8.
  2. Gluzman, Y.; Sambrook, J.F.; Frisque, R.J. Expression of early genes of origin-defective mutants of simian virus 40. Proceedings of the National Academy of Sciences 1980, 77, 3898-3902, doi:doi:10.1073/pnas.77.7.3898.

Round 2

Reviewer 1 Report

Comments and Suggestions for Authors
  1. The Title: The title emphasizes the application of the system for antiviral drug screening. However, this study didn’t include any antiviral data. So, the authors should either modify the title or add some antiviral data

We have modified the title accordingly.

In lines 20–21: “This work describes the establishment of a virus-secreting cell line as an essential step in drug screening methods”, yet the manuscript contains no drug screening experiments or validation. Remove or rephrase this sentence to match the actual scope of the study.

  1. Viral Genome Heterogeneity
  • The study identifies multiple BKPyV variants with structural rearrangements (e.g., NCCR inversions) in COSSA cells. It remains unclear how these variants coexist—do they compete, complement, or act independently?

The manuscript reports multiple integration forms, but it is unclear whether these co-exist in the same cell or derive from distinct subpopulations. Clarify this point or acknowledge it as a limitation if single-cell resolution is not available.

  1. Data Presentation, Figures, and Statistical Rigor

The overall figure layout and content require reorganization to enhance logical flow and highlight key findings. Please revise:

  • All figures must include essential elements such as informative titles, axis labels (with units, e.g., for qPCR in Figure 1), legends, and scale bars where applicable. Quantitative experiments (e.g., FACS, qPCR) lack detailed statistical reporting. Please indicate the number of biological and technical replicates, display error bars, and provide p-values with corresponding statistical tests (e.g., Figures 3B and 4).

All figures presented include titles in the legend head line, statistical analyses and are based on experiments performed in triplicate or quadruplicate to ensure reproducibility and reliability of the data. Number of biological replicates are indicated in the figure legend.

  • The number of figures is excessive, and many contain only minimal data. This reviewer recommend consolidating related figures into multi-panel composite figures and limiting the number of main figures to 3–6. For example:
  • Figure 1: Characterization of the COSSA cell line.
  • Figure 2: Infection of tubular progenitor cells by virus secreted from COSSA
  • Figure 3: ONT sequencing results showing BKPyV genome integration into host cells
  • Figure 4: Structural rearrangements and mutation interpretation of the BKPyV genome
  • Intermediate data such as Figure 11 could be moved to the supplementary materials.
  • Additionally, please ensure that all quantitative figures include complete axis labels, units, sample sizes (n-values), p-values, and descriptions of statistical methods. For instance:
  • Figure 2: Y-axis should indicate “cell count” or “VP1 fluorescence intensity.”
  • Figure 5: Y-axis should indicate “BKPyV copies in supernatant (copies/mL).”
  • Statistical annotations should follow standard reporting format, such as: “Data are presented as mean ± SD/SEM. Statistical analysis was performed using a two-sided paired Student’s t-test (n = XXX, P = XXX, indicated by **).”
  • Figure3:cell cycle analysis should follow standard flow cytometry presentation, including G0/G1, S, and G2/M gating and phase percentages.
  • Figures 4–6 show infection of tubular progenitor cells by virus secreted from COSSA cells. The infection rate should be quantified to correspond with the viral load measured in the supernatant. Since BKPyV assembles in the nucleus, VP1 and LTAg staining typically co-localize with DAPI, which appears inconsistent in this manuscript. The authors should provide an explanation or interpretation for this result.

  1. Misleading or Unsupported Statements
  • Line 265: The claim that LTAg expression is “downregulated due to virion formation” is not sufficiently supported. LTAg is typically expressed early and then declines as replication progresses. Either provide supporting literature or revise the statement to reflect known viral replication dynamics.

We agree with the reviewer’s comment and have revised the descriptive statement to reflect that cells exhibiting high VP1 expression show reduced LTag staining intensity. No further conclusion is taken from this.

Line 623–625:“...display a concomitant downregulation of LTAg expression from the integrated SV40 origin...” suggest LTAg is downregulated due to virion formation, which is not well supported, rephrase or remove this statement unless supported by relevant references.

Line 373 “uper” should be revised to upper.

The rest questions have been addressed.

Author Response

Dear Reviewer, please find the response in the file attached 

Reviewer 2 Report

Comments and Suggestions for Authors

I believe the authors have implemented several positive changes; however, there are still significant issues that require attention. The manuscript contains numerous errors that suggest it was not thoroughly reviewed prior to submission. For instance, there are inconsistencies in figure numbering, and some images are referenced out of order. Additionally, several of the previously suggested revisions—particularly those related to data interpretation and presentation—have not been adequately addressed. I have compiled a list of specific concerns in the attached Word document. 

Author Response

Dear Reviewer, please find the response attached 

Round 3

Reviewer 1 Report

Comments and Suggestions for Authors

Figures 4-6 merge into a single composite figure is recommend.

Author Response

Dear Reviewer, please see the response in the attached file

Reviewer 2 Report

Comments and Suggestions for Authors

General Comments and Corrections:

  1. Lines 297–301:
    The statement regarding differences in cell cycle status is not accurate here in this part. Please remove the entire section from lines 297 to 301.

  2. Figure 2:

    • The figure has been modified to clearly say that there n VP1-positive and VP1-negative cells:   for example:    The mean fluorescence intensity reflects the VP1 signal in COSSA cells, but not in COS cells.
    • FIgure 2c: 
    • The axes in panel C are not clearly explained (MEAN INTENSITY fluorescence?)
    • How can the mean fluorescence of COS cells be 0 if the graph (B) shows a value around 100? Please clarify this in the figure legend  (including a detailed explanation of what data is being displayed)
  3. Marker Description:

    • Line 368: Please describe aquaporin-1 (AQP1) as a membrane marker.
  4. Figures 3–6 and Supplementary Figures 4–6:

    • The resolution is too low — the text is unreadable. This is especially critical for Figures 3–6, which contain essential data. Please provide high-resolution versions of these figures to ensure clarity and readability.

Author Response

Dear reviewer, please see the response in the attached file.
